# Incidence of Prostate Cancer in Inflammatory Bowel Disease: A Meta-Analysis

**DOI:** 10.3390/medicina56060285

**Published:** 2020-06-11

**Authors:** Edoardo Carli, Gian Paolo Caviglia, Rinaldo Pellicano, Sharmila Fagoonee, Stefano Rizza, Marco Astegiano, Giorgio Maria Saracco, Davide Giuseppe Ribaldone

**Affiliations:** 1Unit of Gastroenterology, Department of Medical Sciences, University of Turin, 10126 Turin, Italy; stefanorizza@live.it (S.R.); giorgiomaria.saracco@unito.it (G.M.S.); davidegiuseppe.ribaldone@unito.it (D.G.R.); 2Department of Medical Sciences, University of Turin, 10126 Turin, Italy; gianpaolo.caviglia@unito.it; 3Unit of Gastroenterology, Molinette Hospital, 10126 Turin, Italy; rinaldo_pellican@hotmail.com (R.P.); marcoastegiano58@gmail.com (M.A.); 4Institute of Biostructure and Bioimaging, CNR c/o Molecular Biotechnology Centre, 10126 Turin, Italy; sharmila.fagoonee@unito.it

**Keywords:** Crohn’s disease, ulcerative colitis, urological cancer, screening, systematic review

## Abstract

*Background and objectives:* Inflammatory bowel disease (IBD) is associated with an increased risk of developing colorectal cancer as well as some extra-intestinal tumors, but there are still limited data about the risk of prostate cancer (PC). To analyze if there is an increased risk of PC in patients affected by IBD, we performed a systematic review with meta-analysis. *Materials and Methods:* A Pubmed search of all studies comparing standardized incidence ratio (SIR) or odds ratio (OR) or relative risks (RR) of PC between IBD and non IBD groups, published until March 2020 was conducted. The study protocol was registered on PROSPERO. Twelve studies, mostly population studies, were included. The quality score of these studies, evaluated by the Newcastle–Ottawa Scale, was 7. The heterogeneity was high among the studies in which ulcerative colitis (UC) was considered separate from Crohn’s disease (CD) and in the studies that considered UC and CD together (“IBD-studies”), while it was low in the studies which considered CD separate from UC. *Results:* The relative risk of developing PC was 1.71 (95% confidence interval [CI] 1.16–2.51, *p* = 0.007) in IBD, 1.10 (95%CI 0.98–1.25, *p* = 0.116) in CD, and 1.22 (95%CI 0.98–1.51, *p* = 0.07) in UC. *Conclusions:* Patients with IBD appear to have a slightly increased risk of PC compared to the general population.

## 1. Introduction

Inflammatory bowel disease (IBD) is characterized by a chronic idiopathic gastrointestinal inflammation due to a dysregulated immune response. Based on clinical, histopathological, and endoscopic features, IBD can be differentiated in Crohn’s Disease (CD) and Ulcerative Colitis (UC) [1]. Although it is reported that the pathogenesis of IBD originates from a dysregulated interaction between genetic and environmental factors, the precise etiology is still unknown [2,3]. Patients may also present extra-intestinal manifestations, such as arthritis, erythema nodosum, uveitis, or primary sclerosing cholangitis [4]. It is estimated that 1.5 million Americans and 2.2 million people in Europe are affected by IBD, and several studies have identified an increase in incidence, particularly in newly industrialized countries [5,6].

Chronic inflammation is a well-known risk factor for cancer development, and in the literature, it is widely acknowledged that IBD is associated with a higher risk of intestinal cancers, such as colorectal cancer or small bowel adenocarcinoma [7,8]. Several studies showed that IBD may also be linked to several extra-intestinal neoplasms, such as lymphoma, melanoma, and cholangiocarcinoma [9,10].

Prostate cancer (PC) is the second most frequent malignancy in men worldwide, but only limited and controversial data on the risk of PC in IBD patients are available [11]. Indeed, some studies pointed out an increased risk of PC in IBD, while others did not confirm this conclusion [12,13].

In the present study, we conducted a systematic review with meta-analysis aimed at analyzing the association between IBD and PC.

## 2. Materials and Methods

This systematic review with meta-analysis was conducted according to the PRISMA (Preferred Reporting Items for Systematic Reviews and Meta-Analyses) guidelines [14]. The study protocol was registered on PROSPERO.

Articles published in English about risks of PC in patients with IBD were identified through PubMed (“All databases”) searches using the terms “(inflammatory bowel disease [Title] OR Crohn’s Disease [Title] OR Ulcerative Colitis [Title]) AND prostate [Title] AND cancer [Title]”. The final date of the search was March 30, 2020.

Reference lists from published articles were also employed, such as citing articles on PubMed Central. Titles of these publications and their abstracts were scanned in order to eliminate duplicates and irrelevant articles. The inclusion criteria were: (1) studies comparing the incidence of PC between a group of IBD patients and those without IBD or subjects belonging to the general population; (2) studies reporting standardized incidence ratio (SIR) or odds ratio (OR) or relative risks (RR) and the corresponding 95% confidence intervals (CI); (3) whatever the sample was. The exclusion criteria were: (1) studies which did not estimate the association between IBD and PC risk; (2) studies which did not report the number of PC in the two cohorts; (3) reviews; (4) case reports; (5) meta-analyses.

Two authors (D.G.R. and E.C.) independently reviewed titles and abstracts of references retrieved from the literature search and selected potentially relevant studies. The full-text versions of selected studies were then assessed by the two authors to determine whether the inclusion criteria were satisfied. Differences in opinion were solved by discussion until consensus was reached. If an agreement failed to be reached, a third author (G.M.S.) was consulted.

### 2.1. Statistical Analysis

The following information was collected: first author, country, publication year, number of patients included, numbers of cancers observed in the cohort, expected numbers in a matched background population, SIRs or OR or RRs, and 95%CI.

Cochran’s Q and I2 statistics were used to estimate heterogeneity across studies. When assessing heterogeneity (*Q*-test’s *p* < 0.05, I2 > 50%), a random-effect model was used (this model tends to give a more conservative estimate, i.e., with wider confidence interval), otherwise a fixed-effect model was employed.

*p* value < 0.05 was treated as statistically significant. Statistical analyses were conducted using Med Calc^®^ version 18.9.1 software (Ostend, Belgium).

### 2.2. Quality Assessment and Risk of Bias

The Newcastle–Ottawa Scale (NOS) was used to evaluate the quality of the included studies. The NOS is based on a “star system” and judges study quality according to three perspectives: selection of the study groups, comparability of the groups, and ascertainment of the outcome of interest. The average of quality score was 7 for all the included studies (Table 1).

To identify publication bias, funnel plots were created and visually evaluated. In the absence of publication bias, the distribution of effect estimates in the funnel plot resembles a symmetrical inverted funnel, whereas heterogeneity, publication bias, or chance lead to an asymmetrical funnel plot [24].

## 3. Results

### 3.1. Studies Selected

We identified 150 studies satisfying the search terms. After screening titles and abstracts, 12 of these underwent full-text review. During this process, four papers were excluded as one study was a case-report, one study did not report the exact number of patients with PC, and two studies did not have a comparison with a non-IBD population. By reviewing the lists of references from the retrieved articles, four other articles were included. Twelve studies were finally included, 10 of them were cohort studies, while two had a case-control design. Among these studies, six evaluated the risk of PC in both CD and UC, three evaluated only the correlation between UC and PC, one explored only the correlation between CD and PC, and two referred to IBD patients, without differentiating between the two types [10,12,13,15,16,17,18,19,20,21,22,23].

The flow diagram about the studies’ identification, screening, eligibility, and the number of included studies is reported in Figure 1.

### 3.2. Prostate Cancer in IBD Patients

Among the eight studies evaluating PC risk in IBD (Table 2), three did not show any difference of risk of PC in IBD. The first study was made by Bernstein et al., who investigated the incidence rate ratio (IRR) of various cancers between a cohort of 5529 IBD patients and the general population of Manitoba County. In this study, the IRR for PC was 0.86 (95%IC: 0.57–1.28) [16]. A similar result was obtained by Wilson et al., who compared the risk of cancer between a cohort of 19,647 patients with IBD and an equivalent cohort of subjects without IBD, using data from the clinical practice research datalink (UK), and described a SIR for PC of 1.18 (95%IC 0.85–1.63) [20]. The same conclusion was reached by Jussila et al., who in a nation-wide study in Finland assessed the risk of many malignancies among patients with IBD, reporting a SIR for PC of 0.84 (95%IC 0.73–0.97) [13].

Five studies, instead, evidenced an opposite scenario, pointing out an increased risk of PC in patients with IBD. Kappelman et al., in a Danish population-based study, reported an SIR for PC of 1.21 (IC 1.08–1.35) [10]. The same result was obtained in two Asian studies, by So et al. and Jung et al., in the Hong-Kong and South-Korean general population, respectively, which showed a SIR for PC of 2.03 (95%IC 1.02–4.06) and a SIR for PC of 3.06 (95%IC 1.95–4.80) [21,22]. Moreover, the two case-controls studies showed an increased risk of PC in IBD. Burns et al., by comparing the risk of PC between 1033 patients with IBD and 9306 patients without IBD, who presented at clinics within the Northwestern medicine network (USA), found the highest RR of PC (RR 9.32; 95%IC 5.62–15.46) in IBD patients [12]. Finally, Mosher et al., analyzed 2080 patients affected by IBD and 271,898 controls from the Department of Veteran Affairs in North Texas, and found a RR for PC of 1.71 (95%IC 0.97–3.02) in IBD patients [23].

The meta-analysis indicated that IBD was associated with a RR for PC of 1.71 (IC 1.16–2.51, *p* = 0.007). As a high heterogeneity was detected (*Q* test, *p* < 0.0001; I2 = 93.62%), a random effects model was used (Figure 2).

### 3.3. Prostate Cancer in CD Patients

We found six studies comparing the incidence of PC in CD patients with that of the general population (Table 3). Only one study, performed by Hemminki et al. in a nation-wide population study from Sweden with a cohort of 21,788 CD patients, showed a significant SIR for PC of 1.19 (95%IC 1.02–1.40) [19].

The meta-analysis indicated that CD was associated with a RR for PC of 1.10 (95%IC 0.98–1.25, *p* = 0.12). As a low heterogeneity was detected (*Q* test, *p* = 0.29; I2 = 19.05%), a fixed effects model was used (Figure 3).

### 3.4. Prostate Cancer in UC Patients

Nine studies reporting PC incidence in UC patients were analyzed (Table 4). Five of these did not describe any significant difference between subjects with UC and the general population, while four of these pointed out to the opposite scenario. Karlén et al., in a population study conducted in Stockholm County, showed an SIR of 0.70 (95%IC 0.33–1.47) for PC [15]. In a similar population study, conducted in Copenhagen County, Winther et al. reported an SIR for PC of 0.74 (IC 0.28–1.97) [17]. Hemminki et al., instead, by evaluating the incidence of PC in 27,606 patients with UC compared to the Swedish general population, found an SIR for PC of 1.14 (95%IC 1.01–1.28) [18].

The meta-analysis indicated that UC was associated with a RR for PC of 1.22 (IC 0.98–1.51, *p* = 0.07). High heterogeneity was present (*Q* test, *p* < 0.0001; I2 = 81.86%); thus, a random effects model was used (Figure 4).

### 3.5. Quality Subanalysis

When studies with lower quality, considering a NOS score ≥ 7 as a sign of high quality [25], were excluded from the analysis, the results previously described were confirmed, with an increased RR for PC in patients with IBD, but with a non-statistically relevant increase in risk for PC both in patients with CD and UC (Appendix A).

## 4. Discussion

Several meta-analyses and population studies have shown that IBD could be complicated by intestinal cancers and some studies even provided evidence of an increased risk of some extra-intestinal cancers, such as cholangiocarcinoma or hematological malignancies.

In the literature, there are only few studies examining the risk for PC in patients with IBD, and the relationship between IBD and PC is still controversial. In fact, while some studies showed an increased risk for PC in patients with IBD [12,21], others did not show any correlation between these conditions [13,16].

Recently, two meta-analyses were published by Ge et al. and Chen et al., regarding the risk of PC in IBD patients [26,27]. Ge et al. described a significantly increased risk of PC in patients with IBD (SIR 1.33; 95%IC 1.03–1.71), confirmed in those with UC (SIR 1.58; 95%IC 1.08–2.30), but not in patients with CD (SIR 1.12; 95%IC 0.97–1.31). Chen et al., on the other hand, described, in addition to an increased risk for PC in IBD (RR/OR 1.78; 95%IC 1.32–2.41) and UC (OR/RR 1.76; 95%IC 1.06–2.91), also a greater risk for PC in patients with CD (RR/OR 1.29; 95%IC 1.06–2.91).

Our meta-analysis showed a slightly different result: in fact, while we confirmed an increased risk of PC in IBD patients (RR 1.71; 95%IC 1.16–2.51, *p* = 0.007), we did not evidence a statistically significant increased risk of PC in both UC (RR 1.22; 95%IC 0.98–1.51, *p* = 0.07) and CD (RR 1.10; 95%IC 0.98–1.25, *p* = 0.116). The difference can be explained by the fact that we included the highest number of studies in our analysis. For instance, five studies included in our meta-analysis were absent from the meta-analyses performed by Chen et al. [27] (three on UC and CD; two on UC) and four studies were not included by Ge et al. [26] (two on UC and CD, two on UC). Furthermore, we excluded the study by Jess et al. that was instead included by the other meta-analyses, because it did not satisfy our inclusion criteria; in particular, the exact number of PC developed in the population with IBD was not reported. The three meta-analyses had different designs: for example, when assessing the risk of PC in patients with IBD, we considered only studies which included patients with both CD and UC, while both Ge et al. and Chen et al., also included studies which considered only patients with UC or CD.

Overall, we found that patients with IBD had an increased risk of PC, but subgroup analysis revealed that patients with UC or CD did not have any significant association with PC. Several reasons can explain this contradiction. Regarding UC, the studies that did not show an increased risk of PC had a smaller sample size, and consequently, they were more likely to be subject to beta error. In addition, we included studies from various nations, referring to different populations, with highly dissimilar characteristics. For instance, the higher RR for PC in the Asian studies could be due to the lower incidence of PC in the population: in fact, even a small increase in the number of PC in patients with IBD may give a high increase of the RR for PC in these populations.

Some limitations of the present study are: in the analysis, we did not consider some confounders, that may influence the risk and the incidence of PC, such as family history of PC, screening programs for PC through prostate-specific antigen (PSA), or the fact that the IBD patients could have an increased ascertainment of PC because they are more frequently subjected to rectal examinations due to the characteristics of the disease. Furthermore, we did not consider concomitant medications, such as immunosuppressive drugs, that could increase the risk of developing cancer, decrease ability to eradicate infections, and induce direct DNA impairment. Furthermore, there was a high heterogeneity among the studies, probably due to the different age of the participants and follow-up periods. Another possible bias could be the age of the patients of the included studies. Generally, the majority of those affected by PC are older men, whereas UC and CD affect a younger population. Despite that, the included studies matched case and control for age and showed also an SIR (standardized incidence ratio): hence, this possible confounder was eliminated.

Importantly, our meta-analysis included the highest number of studies published on this subject; we enrolled not only cohort studies, but even case-control ones, and we did a sub-analysis, excluding studies with lower quality, thus confirming the obtained results.

## 5. Conclusions

Our study found that men with IBD have an increased risk of PC, but larger studies are needed to evaluate the risk in the IBD subgroups, especially in patients with UC. Further research should aim at evaluating the impact of concomitant medications on the risk of developing PC. These results could be useful in evaluating the possibility of implementing tailored screening programs for PC in patients with IBD.

## Figures and Tables

**Figure 1 medicina-56-00285-f001:**
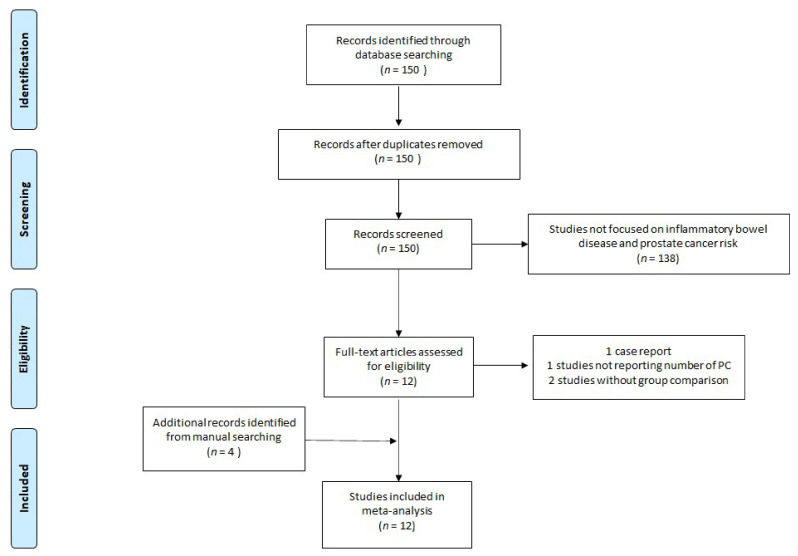
Flow diagram of the study.

**Figure 2 medicina-56-00285-f002:**
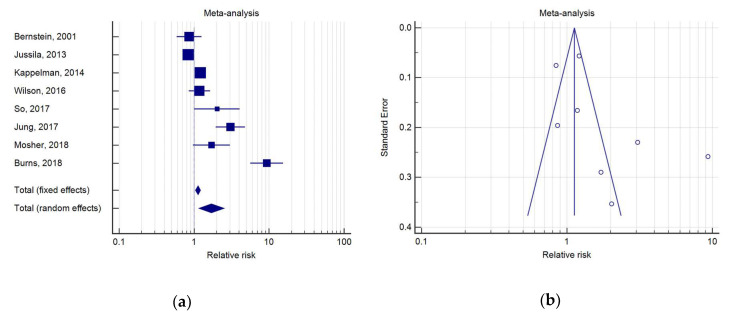
(**a**) Forest plot of relative risks (RR) of prostate cancer (PC) in inflammatory bowel disease (IBD); (**b**) Begg’s funnel plot for IBD.

**Figure 3 medicina-56-00285-f003:**
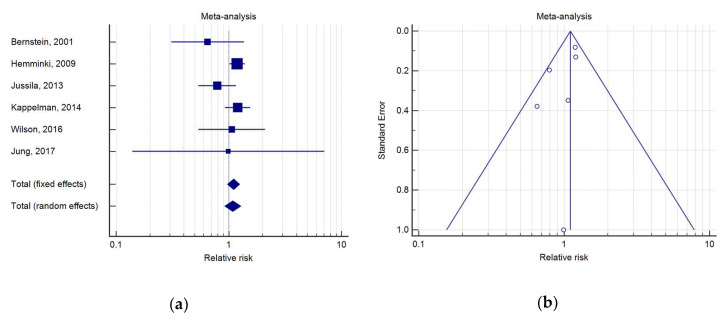
(**a**) Forest plot of RR of PC in Crohn’s disease (CD); (**b**) Begg’s funnel plot for CD.

**Figure 4 medicina-56-00285-f004:**
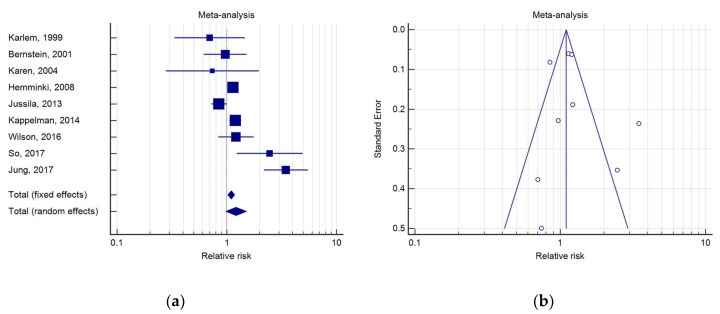
(**a**) Forest plot of RR of PC in UC; (**b**) Begg’s funnel plot for UC.

**Table 1 medicina-56-00285-t001:** Newcastle-Ottawa Scale for the risk of bias assessment of included studies.

Studies	Selection	Comparability	Outcome	Overall Quality Score
*Cohort study*				
Karlen (1999) [15]	4	0	1	5
Bernstein (2001) [16]	4	1	3	8
Whinter (2004) [17]	4	0	2	6
Hemminki (2008) [18]	4	1	2	7
Hemminki (2009) [19]	4	1	2	7
Jussila (2013) [13]	4	0	3	7
Kappelman (2014) [10]	4	1	3	8
Wilson (2016) [20]	4	2	2	8
Jung (2017) [21]	4	1	1	6
So (2017) [22]	4	2	3	9
*Case-control study*				
Burns (2018) [12]	4	2	2	8
Mosher (2018) [23]	4	0	2	6

**Table 2 medicina-56-00285-t002:** Characteristics of the studies about IBD patients included in the meta-analysis.

Author	Location	Region	Data Source	Year	PC/Patients with IBD	PC/Control Population	RR	95%CI
Bernstein et al. [16]	Canada	Manitoba	C.R.	2001	26/5526	6293/1,151,000	0.86	0.59–1.26
Jussila et al. [13]	Finland	Finland	C.R.	2013	176/21,964	51,045/5,351,000	0.84	0.73–0.97
Kappelman et al. [10]	Denmark	Denmark	C.R.	2014	316/42,717	33,960/5,554,844	1.21	1.08–1.35
Wilson et al. [20]	UK	UK	CPRD	2016	79/19,647	67/19,647	1.18	0.85–1.63
Jung et al. [21]	Korea	Korea	HIRA-d	2017	19/15,291	20,607/50,750,000	3.06	1.95–4.80
So et al. [22]	H.K.	H.K.	IBD-d	2018	8/2621	11,115/7,392,000	2.03	1.03–4.06
Mosher et al. [23]	USA	North Texas	C.C.	2018	12/2080	916/271,898	1.71	0.97–3.02
Burns et al. [12]	USA	Northwestern	NMADR	2018	30/1033	29/9306	9.32	5.62–15.46

H.K.: Hong Kong; USA: United States of America; C.R.: cancer register; CPRD: clinical practice research Datalink; HIRA-d: Health Insurance and Review Agency database; PC: prostate cancer; IBD: inflammatory bowel disease; C.C.: case-control; NMADR: Northwestern medicine administrative dataset repository; RR: relative risk; CI: confidence intervals.

**Table 3 medicina-56-00285-t003:** Characteristics of the studies about CD patients included in the meta-analysis.

Author	Location	Region	Data source	Year	PC/Patients with CD	PC/Control Population	RR	95%CI
Bernstein et al. [16]	Canada	Manitoba	C.R.	2001	7/2857	4338/1,151,000	0.65	0.31–1.36
Hemminki et al. [19]	Sweden	Sweden	C.R.	2009	152/21,788	52,621/8,976,000	1.20	1.02–1.40
Jussila et al. [13]	Finland	Finland	C.R.	2013	26/5315	33,134/5,351,000	0.79	0.54–1.14
Kappelman et al. [10]	Denmark	Denmark	C.R.	2014	58/13,756	19,518/5,554,844	1.20	0.93–1.55
Wilson et al. [20]	UK	UK	CPRD	2016	79/19,647	67/19,647	1.18	0.85–1.63
Jung et al. [21]	Korea	Korea	HIRA-d	2017	19/15,291	20,607/50,750,000	3.06	1.95–4.80

C.R.: cancer register; CPRD: clinical practice research Datalink; HIRA-d: Health Insurance and Review Agency database; PC: prostate cancer; CD: Crohn’s disease; RR: relative risk; CI: confidence intervals.

**Table 4 medicina-56-00285-t004:** Characteristics of the studies about UC patients included in the meta-analysis.

Author	Year	Location	Region	Data Source	n. of PC/Patients with UC	n. of PC/Control Population	RR	95%CI
Karlèn et al. [15]	1999	Sweden	Stockholm	C.R.	7/1547	11,526/1,783,000	0.70	0.33–1.47
Bernstein et al. [16]	2001	Canada	Manitoba	C.R.	19/2672	8438/1,151,000	0.97	0.62–1.52
Winther et al. [17]	2004	Denmark	Copenhagen	P.B.C.	4/1160	2598/557,500	0.74	0.28–1.94
Hemminki et al. [18]	2008	Sweden	Sweden	C.R.	277/27,606	79,005/8,976,000	1.14	1.01–1.28
Jussila et al. [13]	2013	Finland	Finland	C.R.	150/16,649	56,718/5,351,000	0.85	0.73–1.00
Kappelman et al. [10]	2014	Denmark	Denmark	C.R.	258/35,152	33,976/5,554,844	1.21	1.08–1.35
Wilson et al. [20]	2016	UK	UK	CPRD	62/11,797	51/11,797	1.21	0.84–1.76
Jung et al. [21]	2017	Korea	Korea	HIRA-d	18/9787	26,904/50,750,000	3.47	2.19–5.51
So et al. [22]	2017	H.K.	H.K.	IBD-d	8/1603	14,936/7,392,000	2.47	1.24–4.93

UK: United Kingdom; H.K.: Hong Kong; C.R.: cancer register; P.B.C.: population-based cohort; CPRD: clinical practice research Datalink; HIRA-d: Health Insurance and Review Agency database; IBD: inflammatory bowel disease; n= number; PC: prostate cancer; UC: ulcerative colitis; RR: relative risk; CI: confidence intervals.

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
