# Peer review of "Incidence of Prostate Cancer in Inflammatory Bowel Disease: A Meta-Analysis"

_medicina, 2020, doi:10.3390/medicina56060285_

Round 1

Reviewer 1 Report

The following are my comments and recommendations for Drs. Edoardo Carli and Gian Paolo Caviglia, on their paper on ‘Incidence of prostate cancer in inflammatory bowel disease: a meta-analysis’.

These results were interesting, and manuscript was written well. But I have concern about discussion the difference between your results and previous meta-analyses. Please see my comments as follows.

  1. I recommend considering confounder of age in included studies. Generally, major prostate cancer population was old age man, whereas UC and CD were young age population. Authors need to evaluate in age population between your included studies and previous meta-analyses included studies. And, please add to discuss this point in the revised manuscript.
  2. I recommend including Table S1 in main manuscript as Table 1.

Author Response

"These results were interesting, and manuscript was written well."

Dear Reviewer, thank you for appreciating our paper.

Q1. I recommend considering confounder of age in included studies. Generally, major prostate cancer population was old age man, whereas UC and CD were young age population. Authors need to evaluate in age population between your included studies and previous meta-analyses included studies. And, please add to discuss this point in the revised manuscript.

A1. Thank you for your suggestion, we added the discussion in the limitation of the study. We considered that, even though there is a discrepancy of age between patients with prostate cancer (old men) and patients with IBD (youngsters), the included studies matched cases and controls for age, eliminating probably this confounder.

Q2. I recommend including Table S1 in main manuscript as Table 1.

A2. We included Table S1 in the main text according to your suggestion.

Reviewer 2 Report

I think the paper was well done and addresses a very important point that has has largely gone unappreciated. I applaud the authors on their approach and work. There are a few corrections that should be addressed prior to publication:

1.  Abstract. 

"The heterogeneity was high in studies on ulcerative colitis (UC) and IBD, while it was low in studies regarding Crohn's Disease. "

I was initially confused by this as UC and CD are a form of IBD and I found this sentence to be very awkward as well as question if i wanted to read any further as I thought there may be some general confusion on IBD. I now understand that some of the papers you reviewed addressed IBD collectively, while others UC and CD apart. As such you did the same which is useful. However, you need to modify this sentence and make this approach more explicit in the paper so readers understand what you mean when you make comparisons of IBD to UC or CD.

2. Please adopt the use of the oxford comma throughout your paper. 

3. Page 2, line 47, "...pointed out to an increased.." I suspect you mean "...pointed out an increased..."

4. Statistical Analysis. "When assessing heterogeneity, a random effect model was used, otherwise a fixed-effect model was employed." Please explain why you chose one or the other in your methods. 

5. Section 3.1 Prostate Cancer in IBD Patients

"Among the 8 studies evaluating PC risk in IBD, 3 did not evidence...". I think you meant to insert the word show, reveal, demonstrate etc. 

6. Throughout the paper, numbers are represented as 5.529 rather than commas, such as 5,529. Check with the journal regarding their preference but I see the comma almost always.

7. You found an increased risk for IBD but not for UC or CD? Presumably, the larger sample size afforded you the power to find statistical significance. I think you should spend more time discussing this in the discussion. 

8. You should also address the differences in the studies (location, region, variations) a bit more thoroughly. For instance, when going over the IBD studies you site 3 that did not have an effect (Canada, UK, Finland) and 5 that did (Danish, 2 Asian studies, 2 USA). I think this is very important and needs more attention to the discussion and analysis. 

Please address these changes. I think that this will be a very useful and impactful paper in clinical practice. 

Author Response

"I think the paper was well done and addresses a very important point that has has largely gone unappreciated. I applaud the authors on their approach and work. There are a few corrections that should be addressed prior to publication."

Dear Reviewer, thank you for your comment and to have evaluated our study.

 Q1. Abstract.

"The heterogeneity was high in studies on ulcerative colitis (UC) and IBD, while it was low in studies regarding Crohn's Disease. "

I was initially confused by this as UC and CD are a form of IBD and I found this sentence to be very awkward as well as question if i wanted to read any further as I thought there may be some general confusion on IBD. I now understand that some of the papers you reviewed addressed IBD collectively, while others UC and CD apart. As such you did the same which is useful. However, you need to modify this sentence and make this approach more explicit in the paper so readers understand what you mean when you make comparisons of IBD to UC or CD.

A1. Dear reviewer, we have rewritten this sentence to eliminate the ambiguity as below:

“The heterogeneity was high in the studies in which ulcerative colitis (UC) was considered separated by Crohn’s disease (CD) and in the studies that considered together UC and CD (“IBD-studies”), while it was low in the studies which considered CD separated from UC.”

Q2. Please adopt the use of the oxford comma throughout your paper.

A2. We added a comma placed immediately after the penultimate term in a series of three or more terms

Q3. Page 2, line 47, "...pointed out to an increased.." I suspect you mean "...pointed out an increased..."

A3. Thank you. We corrected the error.

Q4. Statistical Analysis. "When assessing heterogeneity, a random effect model was used, otherwise a fixed-effect model was employed." Please explain why you chose one or the other in your methods.

A4. We added the explanation that, in case of heterogeneity, the random effect model must be preferred, since this model gives a more conservative estimate.

Q5. Section 3.1 Prostate Cancer in IBD Patients

"Among the 8 studies evaluating PC risk in IBD, 3 did not evidence...". I think you meant to insert the word show, reveal, demonstrate etc.

A5. We corrected with “show”

Q6. Throughout the paper, numbers are represented as 5.529 rather than commas, such as 5,529. Check with the journal regarding their preference but I see the comma almost always.

A6. We change”.” with “,”.

Q7. You found an increased risk for IBD but not for UC or CD? Presumably, the larger sample size afforded you the power to find statistical significance. I think you should spend more time discussing this in the discussion.

A7. Dear reviewer, we agree with you. In fact, we stated in the discussion “Overall, we found that patients with IBD had an increased risk of PC, but subgroup analysis revealed that patients with UC or CD did not have any significant association with PC. Several reasons can explain this contradiction. Regarding UC, the studies that did not show an increased risk of PC had smaller sample size, and consequently, more likely to be subject to beta error. In addition, we included studies from various nations, referring to different populations, with highly dissimilar characteristics. For instance, the higher RR for PC in the Asian studies could be due to the lower incidence of PC in the population: in fact, even a small increase in number of PC in patients with IBD may give a high increase of the RR for PC in these populations.”

Q8. You should also address the differences in the studies (location, region, variations) a bit more thoroughly. For instance, when going over the IBD studies you site 3 that did not have an effect (Canada, UK, Finland) and 5 that did (Danish, 2 Asian studies, 2 USA). I think this is very important and needs more attention to the discussion and analysis.

A8. Thank you for your suggestion. We added locations, region and data source of all the included in the three tables.

We discussed the fact that the higher RR for PC in the Asian studies could be due to the lower incidence of PC in the population: in fact, even a small increase in number of PC in patients with IBD may give a high increase of the RR for PC in these populations. Unfortunately, we cannot perform a sub-analysis on Asian studies due to the too small number of studies.